Research

# Successfully implementing and embedding guidelines to improve the nutrition and growth of preterm infants in neonatal intensive care: a prospective interventional study

Mark J Johnson,[1,2] Alison A Leaf,[1,2] Freya Pearson,[2] Howard W Clark,[2,3] Borislav D Dimitrov,[4] Catherine Pope,[5] Carl R May[5]

Dr Dimitrov died on 15th January 2017, after the manuscript had been finalised but before submission to the journal.

For numbered affiliations see end of article.

**Correspondence to**
Dr Mark J Johnson;
m.johnson@soton.ac.uk

## ABSTRACT

**Objectives** We aimed to improve the nutritional care of preterm infants by developing a complex (multifaceted) intervention intended to translate current evidence into practice. We used the sociological framework of Normalization Process Theory (NPT), to guide implementation in order to embed the new practices into routine care.

**Design** A prospective interventional study with a before and after methodology.

**Participants** Infants <30 weeks gestation or <1500 g at birth.

**Setting** Tertiary neonatal intensive care unit.

**Interventions** The intervention was introduced in phases: phase A (control period, January–August 2011); phase B (partial implementation; improved parenteral and enteral nutrition solutions, nutrition team, education, August–December 2011); phase C (full implementation; guidelines, screening tool, 'nurse champions', January–December 2012); phase D (postimplementation; January–June 2013). Bimonthly audits and staff NPT questionnaires were used to measure guideline compliance and 'normalisation', respectively. NPT Scores were used to guide implementation in real time. Data on nutrient intakes and growth were collected continuously.

**Results** There were 52, 36, 75 and 35 infants in phases A, B, C and D, respectively. Mean guideline compliance exceeded 75% throughout the intervention period, peaking at 85%. Guideline compliance and NPT scores both increased over time, (r=0.92 and 0.15, p<0.03 for both), with a significant linear association between the two (r=0.21, p<0.01). There were significant improvements in daily protein intake and weight gain between birth and discharge in phases B and Ccompared with phase A (p<0.01 for all), which were sustained into phase D.

**Conclusions** NPT and audit results suggest that the intervention was rapidly incorporated into practice, with high guideline compliance and accompanying improvements in protein intake and weight gain. NPT appears to offer an effective way of implementing new practices such that they lead to sustained changes in care. Complex interventions based on current evidence can improve both practice and clinical outcomes.

### Strengths and limitations of this study

► This study was novel in using a sociological theory (Normalization Process Theory) to both guide and measure the process of implementation.
► This study shows that complex interventions, when properly implemented, can change practice in a sustained fashion.
► The before and after methodology used in this study is a limitation and means result should be interpreted with caution, but allowed the implementation process to be studied more closely and in 'real world' conditions.

## BACKGROUND

Attempts to span translational gaps and implement evidence-based practice into routine clinical practice often fail.[1 2] This can mean that patients fail to receive optimal treatment, or conversely may mean they receive unnecessary or potentially harmful care. Neonatal intensive care offers important opportunities for professional behaviour change and practice implementation but is a complex and demanding environment. The neonatal intensive care unit (NICU) has very vulnerable patients with complex and multiple medical problems, and a large multidisciplinary healthcare team working variable shift patterns. It is also a highly technological and information-rich environment. Staff must manage and assimilate a constantly changing array of clinical information from a variety of sources, including monitoring equipment and computerised results systems. It is an interaction-rich environment too: with complex interactions between different professionals, parents and patients themselves. It is a demanding environment to work in, with priorities constantly changing across

the unit as new patients are admitted or others become clinically unstable.

The nutritional care and growth of preterm infants managed in the NICU is an important example of the problem of translating evidence into practice. Recommendations for nutrient intakes have been published,[3] [4] however there is evidence that these are not effectively integrated into clinical practice.[5] There is also evidence that inconsistent and variable nutritional care may be partly responsible for suboptimal growth. Neonatal units offering the same level of care have reported significant variations in rates of postnatal growth restriction and in length of stay, with differences in feeding practices shown to be one of the factors responsible for this variation.[6] Taking this together with the complexity of the NICU environment, it is understandable that current evidence and recommendations for practice fail to be consistently assimilated. We have recently discussed the issues surrounding context and complexity, and it is clear that context has a profound effect on the extent to which new practices can be successfully implemented.[7]

In this paper we describe the successful implementation of a nutrition guideline for preterm infants in a UK NICU leading to sustained change in practice. We show how integrating this guideline into patient care effectively required a carefully designed programme of translational work that facilitated both professional behaviour change (when professionals work differently) and practice implementation (when they embed new ways of conceptualising, enacting and organising practice into their workflow). We explain the operation of this programme of translational work using Normalization Process Theory (NPT),[8] [9] a conceptual toolkit that helped us both to plan guideline implementation and to understand its dynamics.[10] More than 250 studies have now been reported that employ NPT. It offers a rigorous and transferable explanatory model of the mechanisms that promote implementation processes and fits well with the Medical Research Council (MRC) framework for evaluating complex interventions.[11] [12] NPT has four main constructs; coherence (whether people understand the need for change), cognitive participation (whether people understand the change itself and what they need to do to enact new practices), collective action (whether people actually do the work needed for the new practices) and reflexive monitoring (whether people see the benefit of the new practices in their daily work). In figure 1, we show how the mechanisms that drive implementation processes are characterised in NPT. While NPT provides a robust model of implementation that has often been used retrospectively to explain these processes, it has less frequently been used to develop, guide and drive implementation prospectively as it was in the present study.

## METHODS
### Aims
We hypothesised that (1) the implementation of an evidence-based nutrition guideline for preterm infants would improve nutrient intakes and growth; and (2) that the use of NPT to monitor and guide implementation of the guideline would result in its successful integration into

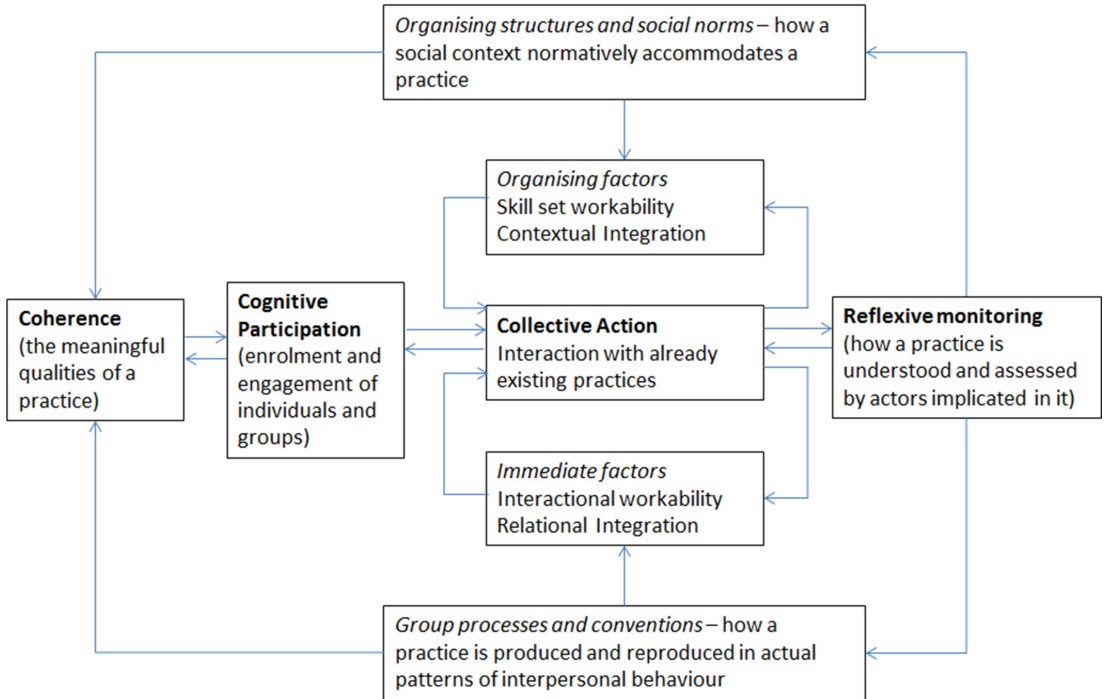

**Figure 1** The model of Normalization Process Theory (NPT). The four main constructs of NPT are shown in bold (adapted from May et al)[8].

practice. We anticipated that improvements in nutrient intake and growth that would follow from successful implementation would have important health benefits.

## Setting and sample

The study was conducted in a NICU in the south of England. Inborn infants with a gestational age less than 30 weeks or birth weight less than 1501 g were eligible for inclusion in the study, and were automatically included from birth to receive the newly implemented service for the provision and monitoring of nutrition for preterm infants. Staff were eligible for inclusion in the study if they were qualified clinicians (nurses, doctors, dietitians) rostered to NICU during phases B, C and D of the implementation study. They took part in individual structured questionnaire (quantitative) data collection using an online tool, and semistructured (qualitative) interviews and focus groups facilitated by MJJ. Figure 2 shows a flow chart of the study.

## Intervention development

A complex intervention was developed with the aim of translating evidence for the nutritional care of preterm infants into practice. It was based on current literature and practice recommendations available at the time (see supplementary additional file 1). To improve the likelihood of successful implementation and embedding in practice, each component of the intervention also aimed to target

implementation mechanisms identified by NPT.[13] The implementation intervention had six major components:

- ► A comprehensive nutrition guideline (see supplementary additional file 1).
- ► A screening tool to identify nutritional risk, linked to specific guideline pathways and nutrition review.[14]
- ► Improved nutritional products: stock parenteral nutrition (PN) solutions were revised to provide more nutrition in a smaller volume and new formula milks and breast milk fortifier introduced with higher nutritional content.
- ► A multidisciplinary nutrition support team (consultant neonatologist with an interest in nutrition, a neonatal dietitian, a neonatal pharmacist and nurse champions).
- ► Nurse champions seconded 1 day in 5 to the nutrition team to improve their knowledge and skills in nutritional care, and 4 days in 5 working clinically, supporting their colleagues in the new ways of working.[15]
- ► A weekly nutrition ward round to review infants at the highest nutritional risk and provide additional management plans for nutrition.

Once developed, the clinical guidelines were circulated to staff and two focus groups held in order to both raise awareness of the changes in practice and to gain insight into potential barriers or facilitators to the

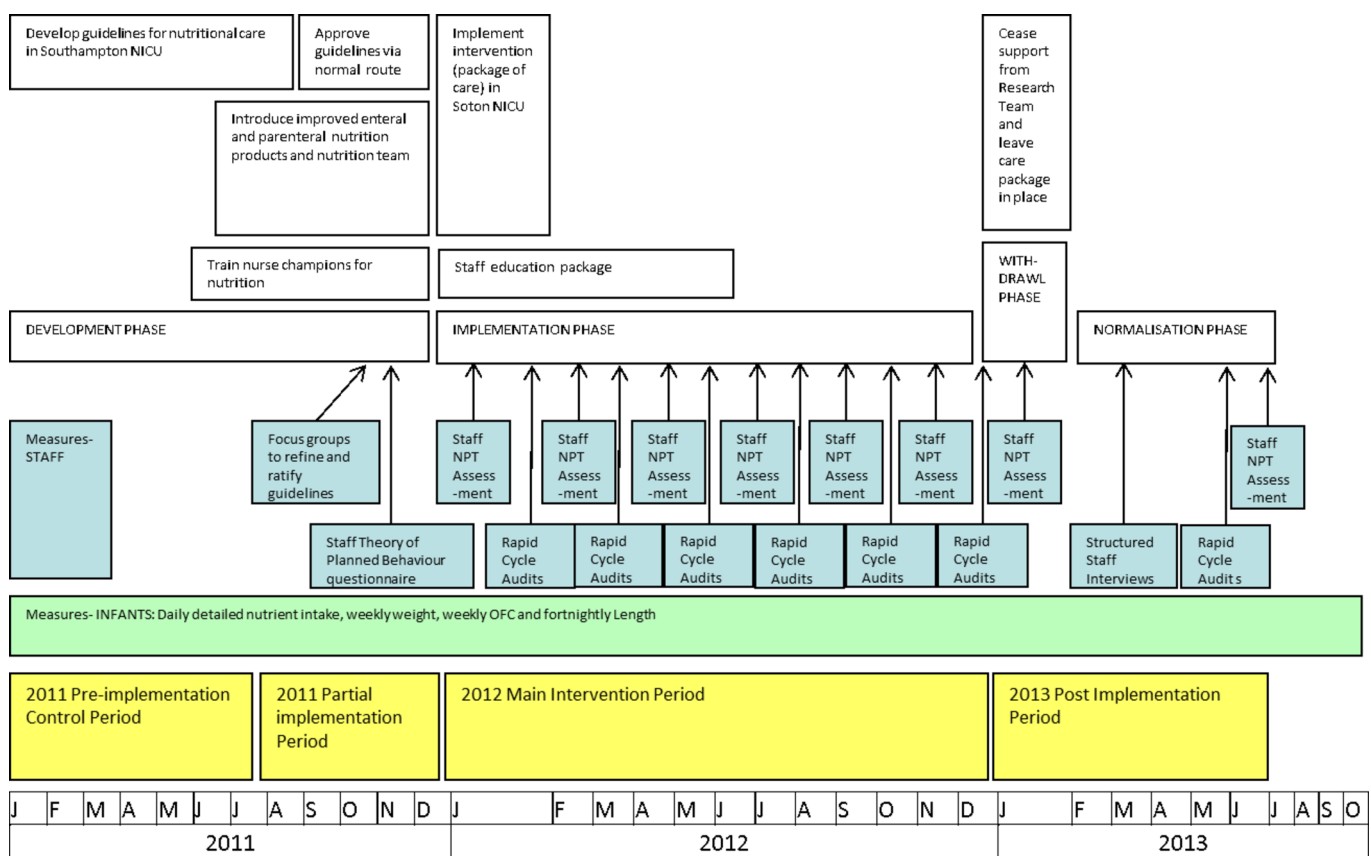

**Figure 2** Study process flow chart. NICU, neonatal intensive care unit; NPT, Normalization Process Theory.

implementation process, enabling tailoring of the guidelines to the local setting.

### Guideline implementation

This was a prospective non-randomized interventional cohort study. Data were collected in discrete periods between January 2011 and June 2013:

A. Control period (1 January 2011 to 31 July 2011). Nutrient intake and growth data on infants born during this period were collected retrospectively after the study had finished in order to provide a contemporaneous 'control' group.

B. Intervention planning and introduction of improved nutritional products (1 August to 31 December 2011). Nutrient intake and growth data on infants were collected prospectively during this period, during which some elements of the intervention (including improved nutritional solutions) were introduced, and staff were consulted about guideline intervention and its associated changes in organisation and practice. In addition, the work with staff carried out during this period to develop the intervention would also be likely to begin to affect practice.

C. Facilitated guideline implementation (1 January to 31 December 2012) during which the full complex intervention was implemented. Nutrient intake and growth data on infants were collected prospectively and audits of guideline compliance and staff NPT toolkit questionnaires were carried out bimonthly.

D. Postimplementation phase (1 January to 30 June 2013). Nutrient intake and growth data on infants were collected prospectively during this period, and one final audit of guideline compliance was carried out to assess the degree to which the new practices remained in place after the main intervention period.

### Patient outcomes

Infant outcomes of primary interest were (1) differences in mean daily energy and protein intakes during stay on NICU between preimplementation and intervention periods, and (2) differences in the change in weight and head circumference (HC) SD scores (SDS) between birth and discharge. These data were collected by entering infant chart data on fluid and feed intake into a specially designed spreadsheet, which was preprogrammed with the nutrient content of feeds and fluids available on the NICU, and automatically calculated daily energy and protein intakes for each infant. Intakes of energy and protein were calculated as raw values and as percentages of the reasonable range of intake (RRI) according to, Tsang *et al*[3] which were the recommendations for the nutritional intake of preterm infants at that time.[3] Of note, these have since been updated by Koletzko *et al* in 2014, which recommends a slightly higher range of energy intake (110–30 kcal/kg/day compared with Tsang's 110–120 kcal/kg/day) and higher range of protein intake (3.5–4.5 g/kg/day compared with Tsang's 3.0–3.6 g/kg/day).[16] Growth data were collected

in a similar manner and converted to SDS using the LMS growth add in for Microsoft Excel using reference data from the UK-WHO Newborn Infant Close Monitoring growth chart.[17] Growth was measured as the change in SDS (cSDS) between birth and discharge. Differences in patient outcomes were also detected by monitoring routinely collected data on mortality, morbidity (eg, necrotising enterocolitis; chronic lung disease; retinopathy of prematurity; severe intraventricular haemorrhage; late-onset sepsis) and length of stay.

### Guideline normalisation and compliance

Measures of nutritional processes were extracted from patient charts at the time of nutritional data entry: time of starting enteral feeds, time of starting parenteral nutrition, time of starting breast milk fortifier and type of feed at discharge. Audits of compliance with the nutrition guideline were carried out throughout the full implementation period, and again at the end of the postimplementation period.[18] Audits were carried out every 2 months in the implementation phase, and once in the postimplementation phase. Measures of the normalisation of guideline compliance were made using a questionnaire based on the NPT online toolkit (www.normalizationprocess.org). This was adapted to ensure that questions related to implementing and embedding the nutrition guideline in practice. This was made available to staff online using www.freeonlinesurveys.com. Respondents were asked to score their level of agreement with each of the 16 items between 1 and 10. This provided overall scores for each of the four domains of NPT (coherence, cognitive participation, collective action and reflexive monitoring). Staff completed questionnaires anonymously.

### Statistical analysis

Descriptive statistics were used to summarise the demographic and outcome variables. The outcome variables were tested for normality using the Kolmogorov–Smirnov test in order to help determine the nature of the analysis methods used, with $p < 0.05$ indicating that the tested variable distribution differed from a normal distribution. For normally distributed continuous variables, the mean and SD were calculated, with the median and IQR calculated for other continuous variables. Distribution of categorical variables was presented as frequency and percentage. Comparison of daily nutrient intake and growth data between periods was carried out using general linear modelling with mixed effects. This statistical technique accounts for repeated measures in the same infant, allowing the addition of other potentially confounding variables (sex, gestational age at birth and birth weight) and subsequent adjustment of the model. Post hoc Tukey's test was used to adjust significance values in view of multiple comparisons. For normally distributed data, a type of general linear model was used, while for non-normally distributed data a type of generalised linear model was used

**Table 1** Infant characteristics in each study group (SD)

| Period | n | Male (%) | Mean birth weight (SD) | Mean gestational age (SD) | Mean CRIB II (SD), n |
|---|---|---|---|---|---|
| A. Preimplementation period (January 2011 to July 2011) | 52 | 23 (44.2) | 1.084 (0.270) | 29.2 (2.6) | 7.0 (3.6), 30 |
| B. Partial implementation period (August to December 2011) | 36 | 18 (50) | 1.029 (0.311) | 29.2 (2.9) | 6.4 (3.9), 20 |
| C. Main intervention period (January to December 2012) | 75 | 37 (49.3) | 0.998 (0.269) | 28.7 (3.0) | 6.9 (2.5), 44 |
| D. Postimplementation period (January to June 2013) | 35 | 22 (62.9) | 0.924 (0.261) | 28.1 (2.8) | 9.7 (3.2), 18 |
| p Value for difference between groups (ANOVA) | | 0.392* | 0.066 | 0.290 | **0.008** |

*p value is for $\chi^2$.

in which repeated measures effects were considered random effects. Missing data were left as missing and not imputed.

Mortality and morbidity data and other dichotomous outcomes were compared across study periods using $X^2$ tests (or Fisher's exact test where numbers were low). Continuous process outcome measures were compared across study periods using either a two-way analysis of variance (for normally distributed data) or the Kruskal-Wallis test (for non-normally distributed data). If statistically significant differences were found then comparisons between pairs of groups were further analysed with post hoc adjustment by Tukey's test (normally distributed data) or multiple Mann-Whitney U tests (non-normally distributed data).

Guideline compliance audit results and measures of the 'normalisation' of practice (using scores from the online NPT questionnaire) were summarised as mean scores and plotted over time. Multiple linear regression was used to describe the nature of the relationship between mean percentage audit compliance and NPT Scores over time. The analyses were carried out using Stata IC V.12.3 (Stata Corp) and SAS V.9.3 (SAS Institute).

## RESULTS
### Measures of Infant Outcomes

Table 1 summarises the sex, gestational age at birth and birth weight of infants in each study period. Clincal Risk Index for Babies (CRIB) II Scores[19] are also shown as an indication of illness severity. CRIB II Scores were not available for all infants and the numbers available with CRIB Scores are also shown in Table 1. There were no significant differences in sex, birth weight or gestational age between groups. There was a significant difference in CRIB II Scores between groups (p=0.008), with post hoc pairwise testing using Tukey's method revealing that only group D was significantly different (higher) from all the others. This suggests an increased level of illness severity in group D when interpreting results.

### Nutrient intakes over time

When compared with baseline data, progressive increases in protein intake were observed over the course of the study. Figure 3A–D shows the results of the generalised linear modelling analysis for mean daily nutrient intakes for each of energy (kcal/kg/day), protein (g/kg/day), energy (as a percentage of RRI) and protein (as a percentage of RRI), respectively, and data tables showing the intake and differences between periods are given in supplementary additional file 2. Using Tukey's test to compare the differences between each period, there were significant improvements in protein intake in periods B and C compared with period A (both p<0.001), and this was sustained beyond the intervention into period D (p<0.01 vs periods A and B). Although there was no significant difference between the partial intervention period (B) and the main intervention period (C) in terms of protein intake, there was a significant increase in protein intake between the partial intervention period (B) and the postimplementation period (D).

### Growth over time

The results of the general linear model using mixed effects for the changes in weight and HC SDS in each study period are shown in figure 4, and data tables showing the intake and differences between periods are given in supplementary additional file 2. Using Tukey's test to compare the differences between each period, there was a significant and sequential improvement in the cSDS from birth for weight in periods B and C compared with period A (both p<0.01), which again were sustained postimplementation in period D (p<0.001 vs periods A and B). There was also a significant improvement in weight between the partial intervention period (B) and the main intervention period (C), suggesting full implementation further added to the intervention effect. This demonstrates that there was a sequential improvement in the difference in weight SDS between birth and discharge in each period during the study. There was a non-significant improvement in the cSDS for HC across the study.

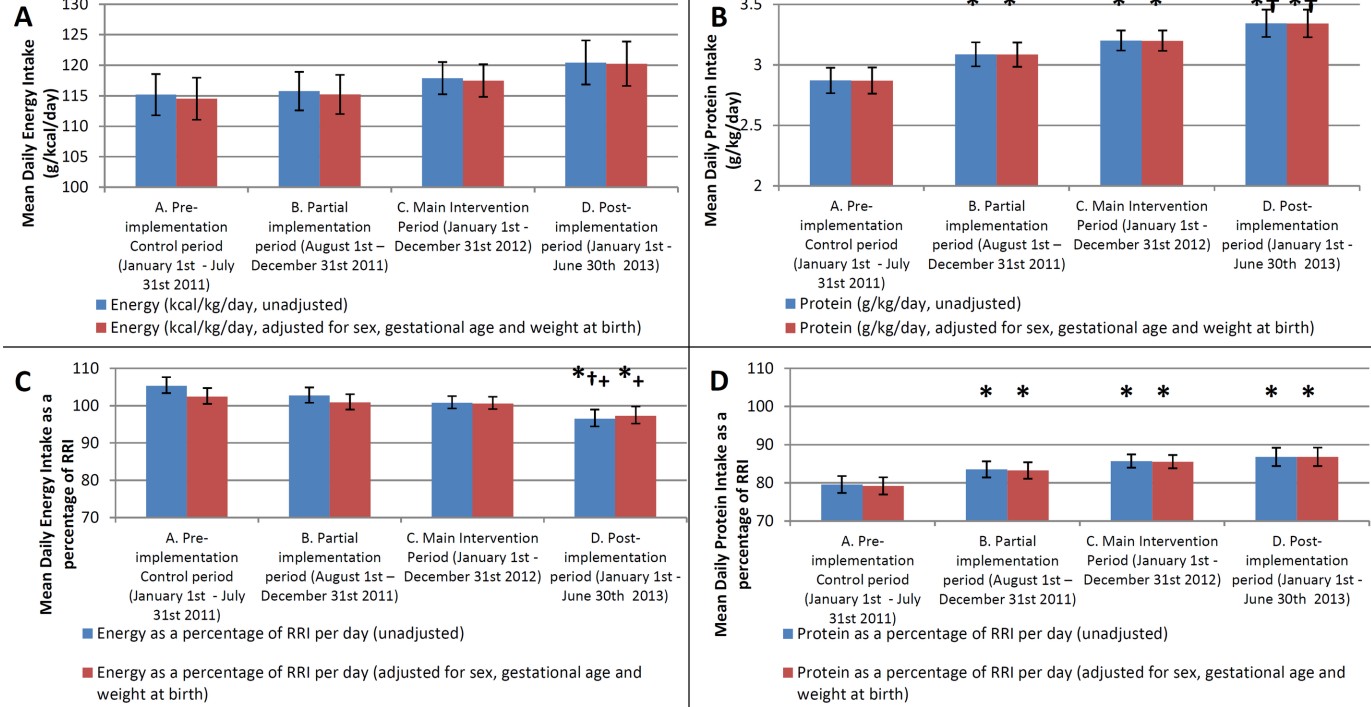

**Figure 3** Bar graphs showing mean daily nutrient intakes across the four study periods. Bars show mean daily energy in kcal/kg/day (A), protein in g/kg/day (B), energy as a percentage of RRI (C) and protein as a percentage of RRI (D). Error bars represent 95% CIs. Blue bars represent unadjusted data, while red bars are adjusted for sex, gestational age and weight at birth. *p<0.05 for difference versus period A, †p<0.05 for difference versus period B,+p<0.05 for difference versus period C. (RRI, reasonable range of intake).

## Mortality and morbidity

No significant differences were detected in the rates of mortality, chronic lung disease, necrotising enterocolitis, severe intraventricular haemorrhage, retinopathy of prematurity and late-onset infection.

## Professional behaviour change and practice implementation

### Timing of commencement of feeds and types of feed

There were no significant differences in the number of babies receiving breast milk, preterm formula, term formula or mixed feeding at discharge between phases of the study. There were no significant changes in the proportion of breast milk fed infants receiving fortifier, nor were there differences in the time to start enteral feeds or the time of starting fortifier in infants receiving breast milk between study periods. However, there were differences in the median time to starting parenteral nutrition between the phases of the study. In the baseline or control phase of the study this was 15 hours. Over the preimplementation and implementation phases of the study this reduced to 9 hours. In the postimplementation phase this rose to 12 hours. A significant difference between study phases was detected using the Kruskal-Wallis test (p=0.013).

### Adherence to guideline

Bimonthly guideline compliance audits—described in figure 2—during the intervention phase and at the end of the postimplementation phase showed that mean compliance improved incrementally across the implementation phase, but there was a slight decrease in compliance at the

final audit in July 2013. Linear regression of mean nutritional audit compliance during the 12 months of the intervention period demonstrated a significant linear increase over time, with a regression coefficient of 1.1 (r=0.92, p=0.009).

### NPT scores

Taking into account participant dropout due to staff turnover, response rates to the NPT toolkit questionnaire peaked at 74% in May 2012, falling to 27% in the final questionnaire in July 2013. Details regarding the number and type of respondents can be seen in Table 2. Figure 5 shows NPT scores as radar plots for each time period; in general, the fuller the radar plot, the greater extent to which staff felt that the practices were part of 'normal practice' at that time. Radar plots generally became fuller over time, though some key areas of the plots were less full at different time points, indicating areas for improvement. The items relating to collective action and reflexive monitoring were scoring lower early in the intervention period, indicating that staff could not see the benefit of the intervention in their work. In order to address this, the results of the study to date were displayed around the staff areas of NICU in August 2012, with a subsequent improvement in the related NPT scores. There was a significant linear increase in mean NPT score over time (coefficient=0.031, r=0.15, p=0.023), though NPT scores fell slightly during the postimplementation phase. Figure 6 shows that global NPT scores and guideline compliance increased together over time

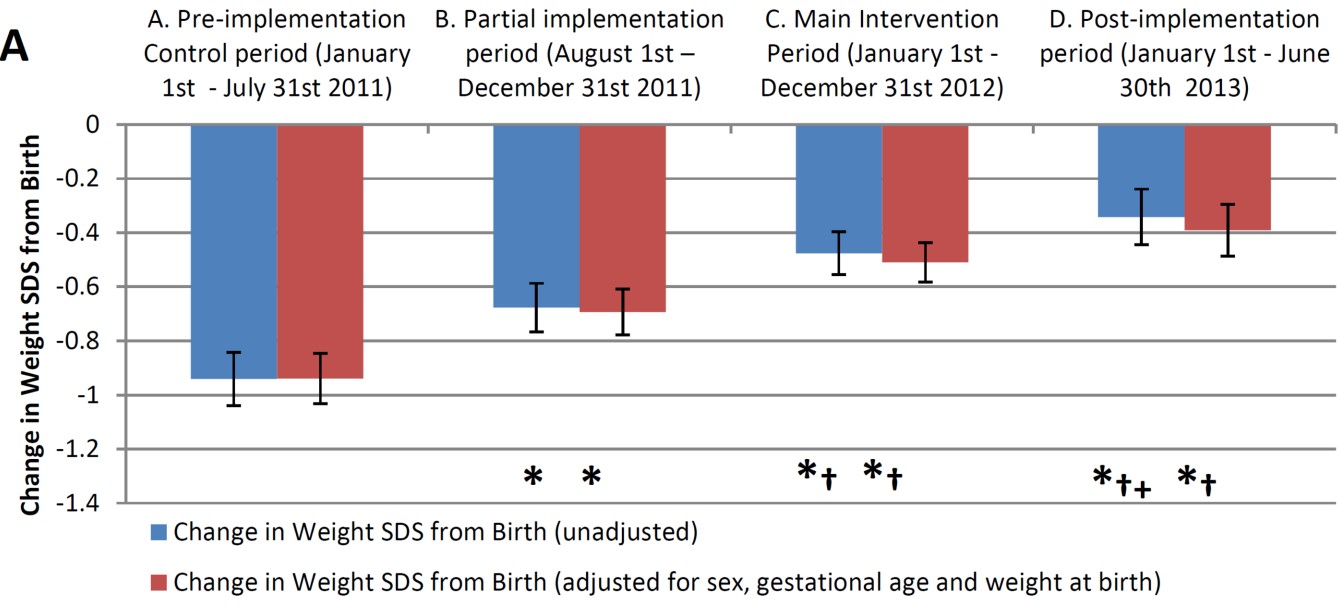

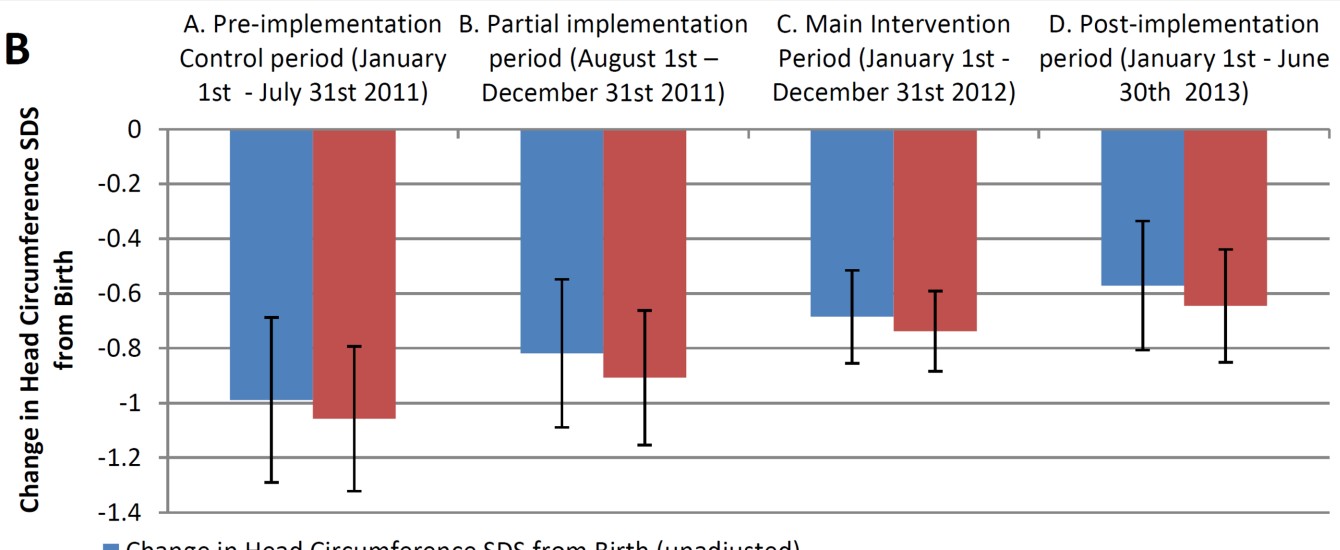

**Figure 4** Bar graphs showing mean change in SD score (SDS) across the four study periods. Changes are from birth until discharge for weight (A) and head circumference (B). Error bars represent 95% CIs. Blue bars represent unadjusted data, while red bars are adjusted for sex, gestational age and weight at birth. *p<0.05 for difference versus period A, †p<0.05 for difference versus period B,+p<0.05 for difference vs period C.

and then flattened out in the postimplementation phase. Linear regression analysis showed that there was a significant association between mean global NPT scores and audit compliance through the intervention development, implementation and postimplementation phases of the study with a coefficient of 0.95 (r=0.21, p=0.002, see Table 3). The addition of time as a variable into the linear regression models (to account for the repeated measures nature of the data) is also shown in Table 3. The addition of time significantly contributed to the increases in compliance over the study and increased the predictive value of the model, though despite this the mean NPT scores remained a significant predictor, showing that the measures of normalisation using NPT are associated with measures of clinical practice.

Linear regression using the mean individual construct scores for NPT showed a significant association with the mean audit scores and participants' capacity to monitor the effects of their actions (*reflexive monitoring*), both before and after adjustment for the effect of time (coefficients of 0.89 and 0.51, p=0.034 and p=0.044 with and without adjustment for time, respectively).

## DISCUSSION

We evaluated the effects of guideline implementation by measuring objective changes in nutrition intake. These data are important in their own right, but can also be used to corroborate subjective self-reports of behaviour change

**Table 2** Number of respondents and percentage response rate for each Normalization Process Theory (NPT) questionnaire

| Time Period | March 2012 | May 2012 | July 2012 | September 2012 | November 2012 | January 2013 | July 2013 |
|---|---|---|---|---|---|---|---|
| Number of respondents | 44 | 52 | 39 | 26 | 24 | 18 | 16 |
| Percentage response rate | 57.9 | 74.3 | 58.2 | 41.3 | 40.7 | 31 | 27 |
| Number (%) consultants | 4 (9.1) | 4 (7.7) | 4 (10.3) | 4 (15.4) | 4 (16.7) | 3 (16.7) | 4 (25) |
| Number (%) junior doctors/ANNPs | 1 (2.3) | 3 (5.8) | 3 (7.7) | 0 (0) | 0 (0) | 0 (0) | 0 (0) |
| Number (%) pharmacists | 1 (2.3) | 1 (1.9) | 1 (2.6) | 0 (0) | 0 (0) | 0 (0) | 0 (0) |
| Number (%) band 7 nurses | 4 (9.1) | 4 (7.7) | 2 (5.1) | 3 (11.5) | 5 (20.8) | 2 (11.1) | 2 (12.5) |
| Number (%) band 6 nurses | 10 (22.7) | 9 (17.3) | 6 (15.4) | 7 (26.9) | 6 (25.0) | 5 (27.8) | 4 (25.0) |
| Number (%) band 5 nurses | 19 (43.1) | 23 (44.2) | 18 (46.2) | 10 (38.5) | 6 (25.0) | 5 (27.8) | 4 (25) |
| Number (%) band 4 nurses | 2 (4.6) | 4 (7.7) | 2 (5.1) | 1 (3.9) | 1 (4.2) | 0 (0) | 2 (12.5) |
| Number (%) band 3 nurses or lower | 3 (6.8) | 4 (7.7) | 3 (7.7) | 1 (3.85) | 2 (8.3) | 1 (5.6) | 1 (6.3) |

ANNP, Advanced neonatal nurse practitioner.

and practice implementation by staff. Objective improvements in nutrient intake and weight gain were detected in infants across the four data collection periods. Against this background, mean audit guideline compliance and NPT scores both increased in a linear fashion over time. Impressively, mean guideline compliance was in excess of 75% throughout the intervention period, peaking at 85%. The headline result of this study is that implementation of the guideline was successfully achieved, and that activities associated with specific intervention components were routinely embedded in workflow within the NICU. This paper has described the successful implementation of a nutrition guideline for preterm infants in NICU, leading to sustained change in practice and improved nutritional outcomes. During the time this study was active, other groups have used similar approaches in the preterm population in order to try and improve infant growth in NICU.[20 21] They also used before and after study designs, but did not include a process evaluation.

Our study has shown that implementing a facilitated nutrition guideline in NICU using a multifaceted intervention improved protein intake and weight gain in preterm infants. Our process evaluation demonstrates that using NPT to develop and guide the implementation process can lead to high compliance with guidelines and changes in practice that are sustained beyond the initial intervention period. The results also show that measures of normalisation using the NPT toolkit correlate well with measures of clinical practice in real life, and suggest that NPT may therefore offer an effective way of measuring and guiding the implementation

process. Effectively implementing the components of this intervention significantly improved both protein intake and weight gain, and appeared to prevent the 'expected' fall of around 1.5–2 SDS for weight between birth and discharge reported in other studies.[22 23] This may be clinically relevant; for example, it may lead to improved neurodevelopmental outcomes[24–26] and so follow-up of the infants in this study will be important. Improvements in weight gain and protein intake appear to continue into the postimplementation period, suggesting that improvements were sustained beyond the main intervention period. It is of interest however, that despite the improvements seen, infants did still fall 0.39 SD for weight between birth and discharge. While such a fall may be considered normal fluctuation around a centile line, it is relevant that even at the end of the study infants still only received around 3.34 g/kg/day of protein (86.8% of RRI) on average across stay, so were still not receiving recommended amounts of protein. This may explain why they still displayed some negative growth. Suboptimal intake of other nutrients such as electrolytes, vitamins and trace elements may also have contributed. Similarly, this may also have contributed to the lack of significant improvements in head growth, although this may in part have been due to poor collection of HC data in the earlier phases of the study (as staff did not begin measuring it consistently until the first intervention period) meaning there were insufficient numbers for a statistically significant result despite a trend towards improvement across the study.

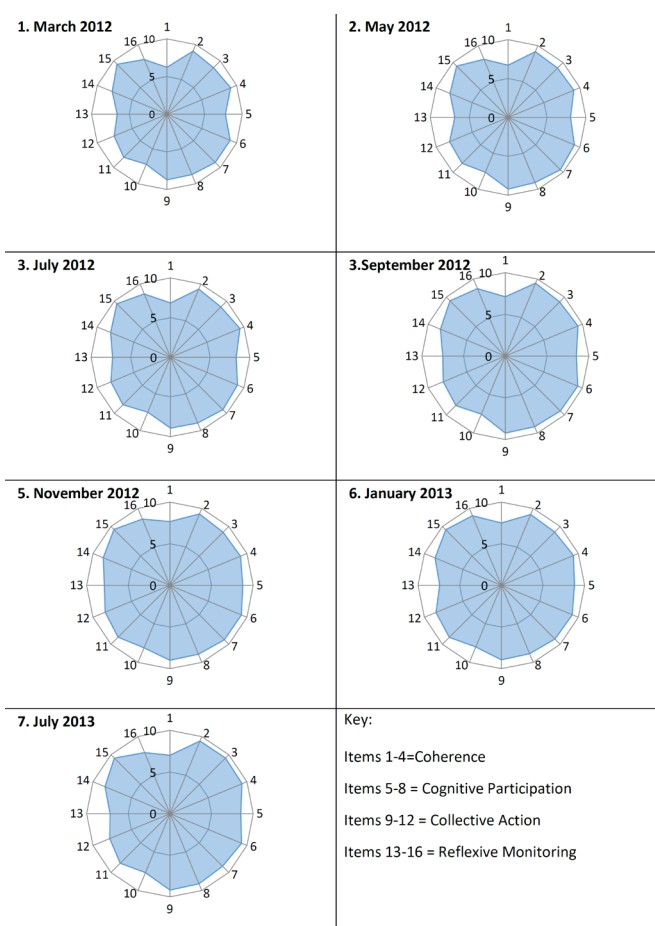

**Figure 5** Radar plots showing the mean score for each subconstruct of Normalization Process Theory (NPT). Results from the NPT questionnaire taken throughout the course of the study.

In the present study, audits of guideline compliance were used in combination with the NPT toolkit. The audits measured how well the guideline was put into practice, and the toolkit provided insight into how well the intervention was being integrated into routine care by staff and identified areas where more work was needed to

**Table 3** Results of linear regression for mean audit compliance measures and mean Normalization Process Theory (NPT) Scores over time

| Outcome | Mean nutritional process audit compliance | |
| --- | --- | --- |
| | Model with time excluded | Model with time included |
| Mean NPT score coefficient (p value) | 0.95 (0.002) | 0.40 (0.031) |
| Time coefficient (p value) | Omitted | 0.72 (<0.0001) |
| p Value for model | 0.0018 | <0.0001 |
| r for model | 0.2098 | 0.8076 |
| $r^2$ for model | 0.044 | 0.6522 |

aid implementation. NPT was used prospectively for the first time in this study to develop and drive the intervention, rather than retrospectively assessing the implementation process. In particular, the guidelines were aimed at encouraging *coherence* and *cognitive participation* by being clear about the reasoning behind the approaches used and how to use them. Similarly, the nutrition team, nurse champions and nutrition ward round aimed to provide feedback to aid *reflexive monitoring*. Audit compliance generally improved over the course of the intervention period, and was around 80%, which is exceptionally high for studies of implementation. NPT scores generally increased over time, suggesting the intervention was becoming 'normalised' into practice. While the use of the NPT toolkit to measure normalisation in this study was novel and experimental, it seems that the measure of 'normalisation' provided by the NPT toolkit does relate to practice changes in the 'real world'. Here, subjective self-reports by staff related well to objective measures of guideline compliance. Global NPT scores were high even at the start of the intervention, suggesting that staff felt the intervention became embedded into routine care rapidly. Importantly, in this study, the use of NPT provided a framework to think through the implementation process,

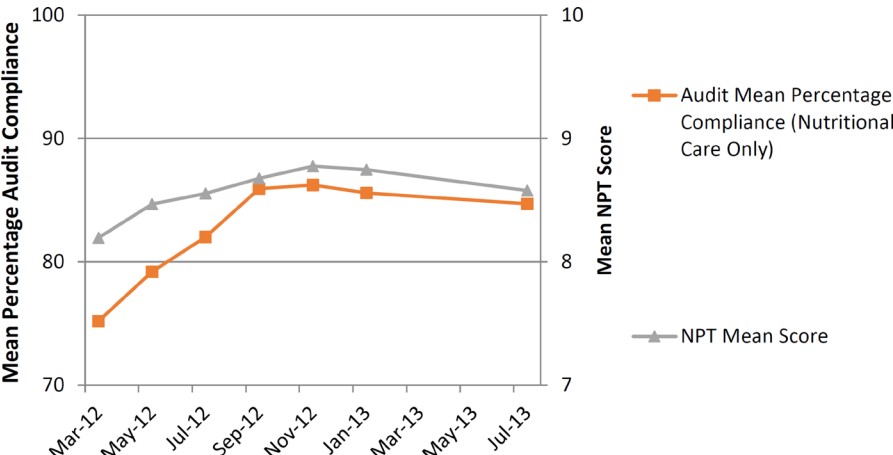

**Figure 6** Relationship over time between mean Normalization Process Theory (NPT) scores and percentage guideline compliance.

with the NPT toolkit measures allowing the implementing team to see where the implementation process could be improved by highlighting how to better engage staff or alter the intervention in areas where NPT scores were low. This unique way of driving, measuring and adjusting the intervention to enhance uptake meant that the use of NPT in this study contributed to the success of the intervention.

A notable result of this study is the importance of reflexive monitoring of implementation progress by staff. This was significantly associated with audit compliance ($r$=0.25). However, it accounted for 6% of the variation in audit compliance and it had an effect size of an improvement of 0.9% audit compliance for every point in global NPT scores. Seeing the impact of personal action functions as a feedback mechanism, and such 'feedback loops' are likely to be responsible for the efficacy of professional interventions such as 'audit and feedback' and 'educational outreach' from other health professionals.[10] Both of these were central components of the intervention. These findings are also consistent with those of a previous theory-led overview of systematic reviews of professional interventions using NPT by our group, which showed that those interventions that emphasised *reflexive monitoring* were more likely to be successful.[10] Showing staff the results of the study to date during the main implementation period in response to low *reflexive monitoring* scores demonstrates the utility of NPT to identify issues and make implementation a more dynamic process. It also illustrates how addressing such issues results in responsive changes that can be seen in subsequent NPT scores, suggesting that NPT offers a way to both *measure* and *guide* change.

We have previously discussed the importance of context in relation to implementation, suggesting that NPT is also able to provide a lens through which to consider the interactions between context and complex interventions.[7] We proposed that the *plasticity* of interventions and the *elasticity* of the context into which they were introduced played a significant part in the degree of implementation success. Using NPT in the present study to both develop and guide the implementation process, perhaps helped overcome the issues with the complex context of the NICU, providing contemporaneous feedback on the barriers to implementation and allowing a degree of plasticity of the intervention itself. This process was also facilitated by the focus groups prior to implementation, allowing potential barriers to be overcome by alterations in the intervention components and the way in which they were delivered. In addition, the focus groups suggested a desire from staff for more consistency in nutritional care, and this in turn is likely to have improved the elasticity of the host context, facilitating *normative restructuring* around the intervention and aiding implementation. This may explain the high degree of compliance and normalisation seen in the present study.

There were some limitations to this study. As a controlled before and after study, it is not possible to be sure if any of the changes seen during the study are a direct result of the intervention. As this was not a randomised controlled trial, it cannot control for causal mechanisms and confounders, and as such it is subject to limits of interpretation. While the statistical analyses show associations between the progressive implementation of the intervention and changes in outcomes, it cannot prove causation. A further limitation relates to having adequate patient numbers and statistical power to detect important differences, which may possibly account for the failure to detect a clinically significant improvement in HC. The study was also not powered to detect differences in mortality and morbidity data. An important limitation of the NPT toolkit questionnaires used in this study is that staff responses may have been biased by their beliefs about the expectations of the study team, which is a common problem in such studies. In addition, the specific interventions used in this study required some additional resources (in terms of the nutrition team) and investment by staff, which may not be available in all units. Several studies have used single interventions such as the introduction of a dietitian or guidelines, and shown improvements in nutrient intakes and growth, without the multifaceted and complex process used in this study.[27–29] While such simple approaches may be more straightforward and require less resource, they are dependent on the expertise of the individuals and their ongoing availability. Our approach employing multiple methods and using sociological theory (NPT) to tailor the intervention to the specific context aimed to embed the changes in nutritional practice into routine care. This enabled it to account for locally available resources, and other units could use a similar approach to develop a multifaceted intervention based on their resources and needs.

## CONCLUSION

This study used nutrition in the NICU as a vehicle to understand implementation in a complex environment. It has demonstrated that the implementation of the facilitated guideline was associated with improvements in infant protein intakes and weight gain. The use of NPT to guide and monitor the implementation of the intervention resulted in high guideline compliance and a degree of 'normalisation' of the complex intervention into routine care. Measures of normalisation using NPT appear to relate to objective measures of practice, suggesting that NPT could provide a useful way of understanding the dynamics of implementation processes in complex clinical environments.

**Author affiliations**
[1]National Institute for Health Research, Southampton Biomedical Research Centre, University Hospital Southampton NHS Foundation Trust and University of Southampton, Southampton, UK
[2]Department of Neonatal Medicine, Princess Anne Hospital, University Hospital Southampton NHS Foundation Trust, Southampton, UK
[3]University Child Health, Faculty of Medicine, University of Southampton, Southampton, Hampshire, UK

[4]Department of Primary Care and Population Sciences, Faculty of Medicine, University of Southampton, Southampton, Hampshire, UK

[5]Faculty of Health Sciences, University of Southampton, Southampton, Hampshire, UK

**Acknowledgements** The authors thank Amanda Beedham and Karen Hayllar at CoEfficient Consultancy, who provided expertise in the development phase and were contracted to develop and build the electronic tool for calculating and storing daily nutrient intake and growth data. The authors also thank Miss Zoe Lansdowne and Amanda Bevan (Neonatal and Paediatric Pharmacists, University Hospital Southampton NHS Foundation Trust) for their help with the parenteral nutrition solution data used in the electronic tool, and Dr Wendy Lawrence for her help with the focus groups. The authors also thank Jenny Pond, Jane Rhodes-Kitson, Charlotte Oates, Jenny Weddell, and Linda Anderson, Christina Humphrey and Liz Blake for their help in data collection and in their roles as nurse 'Champions for Nutrition'. Finally, we would like to dedicate this paper to BDD, who sadly passed away during the publication of this work.

**Contributors** MJJ contributed to the design of the study, carried out data analysis and interpreted all data. He was responsible for drafting the article and revising it critically for important intellectual content. He is guarantor. AAL, FP, HWC contributed to the conception and design of the study and interpretation of data. They revised the article critically for important intellectual content. BDD supervised the statistical analysis and developed the statistical model used for longitudinal data analysis. He contributed to the interpretation of data and revised the article critically for important intellectual content. CP and CRM contributed to the design of the study, the use of NPT in the study and interpretation of data. They revised the article critically for important intellectual content.

**Funding** Work leading to this paper was funded by the National Institute for Health Research (NIHR) Biomedical Research Centre, Southampton. MJJ's contribution was partly supported by an NIHR Doctoral Research Fellowship DRF-2012-05-272, and CRM's contribution was partly supported by NIHR CLAHRC Wessex and partly by ESRC Grant ES-062-23-3274. We gratefully acknowledge the financial support from these agencies. Funders had no role in the design of the study and collection, analysis and interpretation of the data or in writing the manuscript, and the paper does not necessarily represent their views.

**Competing interests** CRM is an original author of Normalization Process Theory; all other authors declare no conflicts of interest.

**Patient consent** Detail has been removed from these case descriptions to ensure anonymity. The editors and reviewers have seen the detailed information available and are satisfied that the information backs up the case the authors are making.

**Ethics approval** The study was approved by an NHS Research Ethics Committee, ('Oxford 'B' Reference 11/sc/0365).

**Provenance and peer review** Not commissioned; externally peer reviewed.

**Data sharing statement** The data sets generated and/or analysed during the current study are not publicly available due to further pending publications and current approvals, but may be available from the corresponding author on reasonable request. An implementation toolkit and a validated instrument to measure implementation processes using Normalisation Process Theory are available at www.normalizationprocess.org.

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
