## [Reviewer comments · BMJ Open]

ARTICLE DETAILS

TITLE (PROVISIONAL)	Successfully implementing and embedding guidelines to improve the nutrition and growth of preterm infants in neonatal intensive care: A prospective interventional study
AUTHORS	Johnson, Mark; Leaf, Alison; Pearson, Freya; Clark, Howard; Dimitrov, Borislav; Pope, Catherine; May, Carl

VERSION 1 – REVIEW

REVIEWER	William W. Hay, Jr., MD University of Colorado School of Medicine USA
REVIEW RETURNED	29-May-2017

GENERAL COMMENTS	The authors report a complex, structured approach to meeting nutritional goals (referenced to Tsang, et al., 2005), but not growth or developmental outcome goals which were not defined or included in the aims of the study, in four groups of VLBW preterm infants over four sequential periods or epochs, showing that nutrition (particularly of protein) and growth (standard anthropometric measurements) were improved over time. While the complex structured approach used by the authors may have benefit among a very mixed group of NICU practitioners (with considerable variability in knowledge base, practice experience, commitment, collaborative capacities, working schedules, etc.), others have done well with a single dietician (+/- physician) who has authority to inform, educate, plan, "order", review, adjust, and so forth. The authors do note that their study has no control group or alternative approach, so whether this approach is better than others remains ambiguous. This is important, because the complex approach used by the authors involves a great deal of work (time, effort, commitment, follow up, etc.), which may not be feasible in many NICUs and, unless markedly better in terms of nutrition, growth, and developmental outcomes, might not be worth all of the effort. Longer term outcomes are likely to correlate with improved nutrition and growth, but this was not assessed in this study, and it does not appear to be a "yet to come" part of the study. Growth parameters, as shown in figure 4, were still negative, even in period 4 (though less so over time, as clearly documented). Why might this be the case? The growth assessments started at birth, not with time of positive weight gain.
---

	Both have merit, but for different reasons, which were not thorough discussed (“from birth” showing responses to earlier and greater amounts of nutrition, “from start of positive weight gain” showing responses to the amount of nutrition and whether this meets the requirements and achieves desired growth outcomes). There was no quantitative assessment of the contribution to improvements in nutrient intakes and growth rates of the four primary components of the nutritional program. Was one more quantitatively important than the others, or two or three combined, or were all equal in their contributions to improved nutrition and growth? This is important to know, as not all NICU/programs might want to put the same effort into all of these components unless they are shown to be fundamental in achieving improved nutrition and growth. Related to the previous point, while there is literature to support Normalization Process Theory (NPT) assessments, it is not clear from the Results/data or the Discussion how much this contributed to the outcomes and whether it should be strongly considered, or not, for incorporation into other NICUs’ nutritional regimens. In other words, what was its unique value and quantitative contribution to outcomes in the current study? Nutrient requirements were based on Tsang, et al. 2005, which has been updated—Koletzko B, Uauy R, Poindexter B, editors. Nutritional Care of Premature Infants. S. Karger AG, 2014. (World Rev Nutr Diet. 2014;110. PMID: 24751622). The authors might want to add a sentence or two to state whether the 2014 updated reference values are different from those in 2005 and whether they are in the direction used for the nutrient regimens in their study. Presumably, the extensive set of documents attached to the manuscript would be made available as part of an appendix or via a weblink. They certainly would be useful for others to review and model their own nutrient regimens after, but too cumbersome for publication within the published journal article.
--	--

REVIEWER	SERGIO VERD BALEARIC HEALTH AUTHORITY, DPT PRIMARY CARE, PEDIATRIC UNIT, Palma de Mallorca, Spain
REVIEW RETURNED	10-Jul-2017

GENERAL COMMENTS	Timing and types of feed ought to be further discussed. The authors report that there were no significant differences in the number of babies receiving breast milk, preterm formula, term formula or mixed feeding at discharge between phases of the study. Their figures show that there is no increase in head circumference, what is in agreement with same proportions of breastmilk across study phases. Are the authors satisfied with these outcomes? On the other hand, we have details on starting parenteral nutrition but why not on withdrawing parenteral nutrition?
---

VERSION 1 – AUTHOR RESPONSE

Reviewer: 1

Reviewer Name: William W. Hay, Jr., MD

Institution and Country: University of Colorado School of Medicine, USA Competing Interests: None declared

Comment

The authors report a complex, structured approach to meeting nutritional goals (referenced to Tsang, et al., 2005), but not growth or developmental outcome goals which were not defined or included in the aims of the study, in four groups of VLBW preterm infants over four sequential periods or epochs, showing that nutrition (particularly of protein) and growth (standard anthropometric measurements) were improved over time.

While the complex structured approach used by the authors may have benefit among a very mixed group of NICU practitioners (with considerable variability in knowledge base, practice experience, commitment, collaborative capacities, working schedules, etc.), others have done well with a single dietician (+/- physician) who has authority to inform, educate, plan, “order”, review, adjust, and so forth. The authors do note that their study has no control group or alternative approach, so whether this approach is better than others remains ambiguous. This is important, because the complex approach used by the authors involves a great deal of work (time, effort, commitment, follow up, etc.), which may not be feasible in many NICUs and, unless markedly better in terms of nutrition, growth, and developmental outcomes, might not be worth all of the effort.

Response:

This is a good point, and worth more explanation in the manuscript, which we have now added. Firstly, whilst single interventions and initiatives such as dedicated dietitians have shown benefit in other studies, these are dependent on the expertise of the individuals and their ongoing availability. Our approach using multiple methods and sociological theory (NPT) was tailored to the specific context and sought to embed the changes in nutritional practice into routine care. This enabled it to account for locally available personnel and resources, and other units could use a similar approach to develop a multifaceted intervention based on their resources and needs. It is likely their intervention would differ from ours, but that is the point of our approach. Furthermore, the guideline was a key element of our intervention, as it empowered staff to make decisions around nutrition even without specific expertise, and this is something that would also be applicable to other units (and we have included this in the supplemental online material). We have highlighted these potential benefits in the discussion, as well as further highlighting the balance between potential benefit and the effort/resource required as limitations (lines 421-430)

Comment

Longer term outcomes are likely to correlate with improved nutrition and growth, but this was not assessed in this study, and it does not appear to be a “yet to come” part of the study.

Response

Neurodevelopmental follow up and growth assessment of these children at 2 years has been carried out, and is currently being analysed. We have added a line to indicate this in the manuscript (line 350)

Comment

Growth parameters, as shown in figure 4, were still negative, even in period 4 (though less so over time, as clearly documented). Why might this be the case?

Response

This is an interesting point. There is a body of evidence that shows that these infants as a group fall down their growth charts, dropping two or more marked centile lines on a growth chart between birth and term equivalent age. Our infants were following this pattern at the start of the study, and by the end were only dropping only 0.39 standard deviations. This is less than one marked centile line on a UK growth chart (0.66 SD) and so could be considered normal fluctuation around a centile line. However, it is also of note that despite our efforts, infants still only received around 3.34g/kg/day of protein (86.8% of RRI) on average across stay, so were still not receiving recommended amounts of protein. This may explain why they still did display some negative growth. Suboptimal intake of other nutrients such as electrolytes, vitamins and trace elements may also have contributed. We have added some text to explain this in the discussion (lines 352-62)

Comment

The growth assessments started at birth, not with time of positive weight gain. Both have merit, but for different reasons, which were not thorough discussed (“from birth” showing responses to earlier and greater amounts of nutrition, “from start of positive weight gain” showing responses to the amount of nutrition and whether this meets the requirements and achieves desired growth outcomes).

Response

This is a good point, and we agree it would be interesting to look at growth from the start of positive weight gain. However, our data does not currently allow us to do this and also we felt that changes in growth from birth are the measure used in similar studies of this type. Furthermore, assessing growth from the point of positive weight gain adds several more layers of complexity, firstly because this will be different for each infant, but will also be influenced heavily by fluid intake rather than just nutritional intake, and this will vary hugely between individual infants. We have now stated change in SDS from birth as our chosen measure of growth in the methods section for clarity (line 193).

Comment

There was no quantitative assessment of the contribution to improvements in nutrient intakes and growth rates of the four primary components of the nutritional program. Was one more quantitatively important than the others, or two or three combined, or were all equal in their contributions to improved nutrition and growth? This is important to know, as not all NICU/programs might want to put the same effort into all of these components unless they are shown to be fundamental in achieving improved nutrition and growth.

Response

The analysis and p values given in the figures refer to differences between each of the four time periods. It is therefore possible to some extent to comment on the effects of the components introduced at each time point, and we have added some text in the results that attempts to highlight the effects at each stage of the intervention (lines 253-4, 257-8, 262-3, 266-8) and have also provided the data showing the size of the intakes/growth and the differences between periods as an supplemental file (additional file 2).

Comment

Related to the previous point, while there is literature to support Normalization Process Theory (NPT) assessments, it is not clear from the Results/data or the Discussion how much this contributed to the outcomes and whether it should be strongly considered, or not, for incorporation into other NICUs' nutritional regimens. In other words, what was its unique value and quantitative contribution to outcomes in the current study?

Response

Thanks for highlighting this- we believe that the use of NPT was an important way of both driving the implementation process and measuring how successfully the intervention was implemented. We have added some additional text in the discussion to further highlight this (lines 378-83). This is also illustrated by the discussion on how important 'reflexive monitoring' was in the success of the intervention (lines 387-400)

Comment

Nutrient requirements were based on Tsang, et al. 2005, which has been updated—Koletzko B, Uauy R, Poindexter B, editors. Nutritional Care of Premature Infants. S. Karger AG, 2014. (World Rev Nutr Diet. 2014;110. PMID: 24751622). The authors might want to add a sentence or two to state whether the 2014 updated reference values are different from those in 2005 and whether they are in the direction used for the nutrient regimens in their study.

Response

This is a helpful comment, and we have indeed added some text in the discussion to highlight the revised recommendations and how they differ to those used in the study (lines 186-190)

Comment

Presumably, the extensive set of documents attached to the manuscript would be made available as part of an appendix or via a weblink. They certainly would be useful for others to review and model their own nutrient regimens after, but too cumbersome for publication within the published journal article.

Response

We completely agree with this comment, and have provided the guidelines materials as online supplemental material (additional file 1)

Reviewer: 2

Reviewer Name: SERGIO VERD

Institution and Country: BALEARIC HEALTH AUTHORITY, DPT PRIMARY CARE, PEDIATRIC UNIT, Palma de Mallorca, Spain Competing Interests: none declared

Comment

Timing and types of feed ought to be further discussed.

Response

Details regarding the timing and types of feed are given in the nutritional guideline used in the study, and this is provided as additional online supplemental material (additional file 1).

Comment

The authors report that there were no significant differences in the number of babies receiving breast milk, preterm formula, term formula or mixed feeding at discharge between phases of the study. Their figures show that there is no increase in head circumference, what is in agreement with same proportions of breastmilk across study phases. Are the authors satisfied with these outcomes?

Response

We did not look specifically at the number of babies who received breast milk at any time, but did look at feed type at discharge. There was a slight increase in breast milk use at discharge home from 40% at the start of the study to 50% at the end, with a concurrent reduction in preterm formula milk use from 35% to 16.7%.

However, using Fisher's exact test across all 4 groups did not show any difference across all four time periods. It may be that pairwise comparison of the two groups at the start and the end may have revealed significant differences but we chose not to do this given that the overall p value for all four groups was non-significant.

The lack of a significant improvement in head circumference was disappointing, though the trend towards improvement was encouraging. It may well be that the poor collection of head circumference data in the earlier phases of the study (as staff did not begin measuring it consistently until the first intervention period) may have contributed to this statistical result, though failing to reach RRI for protein may also have contributed. We have now mentioned this in the text (lines 358-62).

Comment

On the other hand, we have details on starting parenteral nutrition but why not on withdrawing parenteral nutrition?

Response

This is a fair point. However, as our study focussed on implementation, getting PN started earlier was a key audit point. We did not formally collect data on duration of PN, though it is likely that this would have been increased by the intervention, as it changed practice to work towards total fluids targets of 180ml/kg/day before starting to wean PN as feeds were increased, meaning infants would have stayed on PN longer as PN was stopped once 180ml/kg/day feeds were reached (previously we stopped PN at 150ml/kg/day feeds).

VERSION 2 – REVIEW

REVIEWER	Sergio Verd Pediatric Unit, La Vileta Surgery, Department of Primary Care, Balearic Health Authority. Palma de Mallorca, Spain.
REVIEW RETURNED	16-Aug-2017
GENERAL COMMENTS	Satisfactory minor revision required